# Assessing the Effect of Artificial Intelligence Anxiety on Turnover Intention: The Mediating Role of Quiet Quitting in Turkish Small and Medium Enterprises

**DOI:** 10.3390/bs15030249

**Published:** 2025-02-22

**Authors:** Selen Uygungil-Erdogan, Yaşar Şahin, Aşkın İnci Sökmen-Alaca, Onur Oktaysoy, Mustafa Altıntaş, Vurgun Topçuoğlu

**Affiliations:** 1Kadirli Faculty of Applied Sciences, Osmaniye Korkut Ata University, 80750 Osmaniye, Türkiye; suygungilerdogan@osmaniye.edu.tr; 2Beşikdüzü Vocational School, Trabzon University, 61800 Trabzon, Türkiye; yasarsahin@trabzon.edu.tr; 3Faculty of Economics and Administrative Sciences, Istanbul Arel University, 34537 Istanbul, Türkiye; askinsokmen@arel.edu.tr; 4Faculty of Economics and Administrative Sciences, Kafkas University, 36000 Kars, Türkiye; onuroktaysoy@kafkas.edu.tr; 5Çekerek Fuat Oktay Vocational School of Health Services, Yozgat Bozok University, 66100 Yozgat, Türkiye; mustafa.altintas@bozok.edu.tr; 6Faculty of Economics, Administrative and Social Sciences, Istanbul Nişantaşı University, 34398 Istanbul, Türkiye

**Keywords:** artificial intelligence, quiet quitting, turnover intention, management of turnover, SMEs

## Abstract

The concept of artificial intelligence (AI) refers to technologies that imitate human-like thinking, learning and decision-making abilities. While integrating AI into the workforce offers the potential to increase efficiency in organizational activities, it can lead to negative effects such as anxiety, uncertainty, and distrust among employees which results from not being able to understand these technologies, regarding them as alternatives for themselves, and the possibility of losing their organizational position. These effects can reduce employees’ commitment at work and trigger negative organizational behaviors such as quiet quitting and turnover intention. Starting from this point, the present study aims to investigate the effect of AI anxiety on turnover intention and the mediating role of quiet quitting in this relationship. The study was conducted using a cross-sectional design with 457 people working in SMEs in Kırıkkale province. AI Anxiety, Quiet Quitting, and Turnover Intention Scales were utilized during the data collection process. The obtained data were analyzed through structural equation modeling. In addition to detecting significant relationships between concepts as a result of the analysis, it was realized that AI anxiety did not have a considerable effect directly on turnover intention; however, this effect occurred indirectly through quiet quitting. Accordingly, it is predicted that integrating AI technologies into business processes will increase the concerns about job security in employees, and this anxiety triggers the turnover intention by leading to a tendency toward quiet quitting for reasons such as loss of motivation and low organizational commitment.

## 1. Introduction

The concept of AI is defined as a technological structural whole that includes a series of technologies imitating human characteristics such as machine learning, and language and image processing ([31]). As a reflection of technological advancements and the continuous development of AI systems, the area of use for AI is increasing day by day ([48]). With the help of AI applications, it becomes more possible to expand quality, reliable, and cost-effective services. For instance, in the field of healthcare, a 75% success rate was achieved thanks to AI in detecting faulty medications prescribed to patients, and 56.20% of valid warnings were evaluated as critical ([50]). It is found that employees who utilize AI while performing their duties carry out tasks 25% faster and produce 40% higher-quality results ([43]).

The emerging situation reveals that AI applications have a great effect in terms of higher-quality service and reducing costs in many areas. A McKinsey study published in June 2023 indicates that annual economic benefits ranging from USD 2.6 trillion to 4.4 trillion can be achieved in activities covering 16 business functions through 63 different AI applications. Within the scope of the research, it was found that 850 different professions will be affected, and this figure is expected to be USD 15.7 trillion in 2030 ([42]).

AI is depicted in the literature as reducing the workforce to a great extent, although not eliminating it completely. Industrial robots and AI technologies influence the production and marketing processes of businesses significantly through labor substitution and technology updates. It is predicted that AI will replace 54% of jobs in Europe in the next 10–20 years ([72]). However, this situation also raises concerns about AI.

Today, there are many theories that explain the reasons for AI anxiety. In light of the uncanny valley theory ([55]), it is predicted that AI will lead to some negative behaviors and perceptions from the point of employees such as quiet quitting and turnover intention so long as it has human characteristics ([7]). The theory in question focuses on the relationship between the similarity of an object or program to a human being and the closeness of an individual perceiving it to the object or program. In this regard, the uncanny valley theory suggests that as AI outputs possess characteristics that are attributed to humans, anxiety, fear, and various psychological problems may arise in the individual ([12]). The theory is frequently used in popular media to explain people’s rejection of self-similar robots and virtual agents as well as to explain the failure of computer-animated films such as The Polar Express. The use of AI technologies produced to help people or make things easier while completing a certain task or job is considered appropriate for individuals. On the other hand, in cases where the limits of AI that imitates humans and behaves like humans are not determined, individuals tend to reject AI products by demonizing them ([69]).

The way employees perceive AI, the reason why they use it, and the way they establish sociological connections have not been fully explained in the previous literature ([8]). There are serious concerns about integrating AI into business life without fully comprehending its effects on ethics, education, and security ([63]). Many organizations such as the OECD strive to address concerns, establish certain standards, and use AI responsively ([60]). However, studies reveal that AI can result in different consequences in terms of application, even in organizations that do the same job ([9]; [81]). For example, in the study conducted by [51] ([51]), AI is perceived as an “algorithmic counterpart” that makes things easier and helps ensure safety by the Amsterdam police department. However, due to the hierarchical structure of the Berlin police department, AI applications are perceived as an “algorithmic lattice”. The fact that different results are obtained even between the two units performing the policing task indicates that AI is affected by organizational variables to a large extent ([11]). The present study tried to explain the possible quitting and quiet quitting behaviors that may exist with including AI in business life through SMEs in Turkiye. In this respect, this study is expected to contribute to the explanation of the causes and consequences of AI anxiety. It also reveals that quiet quitting, a new concept in the organizational behavior literature ([19]; [71]), along with the behavioral tendency structure within the organization, is also expected to be explained. The integration of turnover intention into the study will help clarify concerns about “technological unemployment” and the “dark side of AI” ([30]). In this regard, it aims to contribute to the literature by making use of the uncanny valley theory, and to explain the turnover intention and quiet quitting behaviors that occur in the organization as a result of AI anxiety.

In this respect, we aim to contribute to the literature by making use of this theory to explain the turnover intention and quiet quitting behaviors emerging in organizations as a result of AI anxiety. This study was conducted cross-sectionally with 457 people working in SMEs, and the data obtained were interpreted through structural equation modeling. Unlike previous studies, the present study examines the negative effects of AI on employees at the organizational level and seeks answers to the following questions:Are AI anxiety, quiet quitting behavior, and turnover intention considered organizationally common behaviors among employees in SMEs?Are the employees within the organizational structure aware of the concepts in question?Does AI anxiety have an impact on quiet quitting behavior and turnover intention?What is the interaction between quiet quitting behavior and turnover intention?Which theories explain the interaction and mediation relationship between variables?

The answers to the questions presented above are expected to make a contribution to the literature.

## 2. Conceptual Framework

### 2.1. AI Anxiety

AI anxiety can be described as a concept expressing individuals’ fear, uncertainty, and negative perceptions towards the future effects of AI technologies. This concern can be based on various reasons such as fear of unemployment due to AI technologies, ethical concerns, violation of privacy, and inability to manage technological development ([2]). In the current world, where technology is developing rapidly and occupies an increasingly larger place in daily life, AI anxiety has become a critical issue for individuals, communities and organizations. This anxiety not only creates psychological discomfort at the individual level but can influence the innovation processes and technology adaptation strategies of organizations as well ([45]).

AI anxiety has a twofold importance for organizations, the first of which is related to the internal processes of the organization. At this point, anxiety about AI among employees brings about the risk of reducing workplace motivation and loyalty. Despite the potential of AI applications in organizations to enable greater efficiency and cost savings in business processes, employees feeling threatened can sometimes jeopardize the success of such initiatives and trigger more negative scenarios in the organizational sense ([80]). Secondly, similar concerns about AI may arise among customers and stakeholders. In particular, concerns about privacy and data security can make it difficult to adopt AI-based products and services as well as destroy the relationship and interaction between the parties ([83]). In this respect, it is of utmost importance for organizations to increase their credibility by using AI in accordance with ethical rules.

When the reasons leading to AI anxiety are considered within the scope of academic research, it can be realized that they are generally categorized into four groups. The reasons in question are as follows:Reasons arising from unemployment and economic insecurity: The potential of AI to reshape the labor market can trigger employees’ fear of losing their jobs. In particular, the prediction that routine and low-skilled jobs will be replaced by AI-based systems with the increase in automation causes uncertainty and stress among employees ([74]). This situation has the risk of increasing not merely the economic security of individuals but also social inequalities.Reasons arising from ethical problems and concerns: AI’s capacity to make autonomous decisions causes ethical dilemmas. For example, algorithmic bias, issues of discrimination and accountability undermine individuals’ trust in AI significantly ([82]). Furthermore, the lack of ethical rules increases concern about the uncontrolled progress of technology.Reasons arising from privacy concerns: The fact that AI is based on big data analytics and requires some personal information causes individuals to experience various concerns about the privacy of their personal data. The fear of privacy violation results in high anxiety, especially in sensitive data areas such as social media, health, and finance, and therefore can make the adoption of AI-based systems difficult ([83]).Concern about loss of control over technology: The continuous development of AI brings about fears that the control of technology over people will increase and that these technologies may even cause harm to human interests. This situation is likely to cause individuals to develop resistance by feeding concerns about dystopian scenarios as well as causing individuals to worry about technological advances ([45]).

It is known that seemingly harmless psychological phenomena underlie many emotional reactions. In this regard, the uncanny valley theory is thought to lead to some undesirable organizational changes with the concern about AI that drags people into the unknown ([54]). As explained above, briefly, AI-based systems are perceived as a threat by employees, and for this reason, negative behavior of employees towards production and the organization is expected.

### 2.2. Turnover Intention

The success and sustainability of businesses are to a large extent related to the long-term stay of talented and experienced employees in organizations. Employee continuity not only ensures the preservation of organizational knowledge but also contributes to reducing costs, increasing the efficiency of business processes and strengthening team dynamics ([4]). At this point, turnover intention is regarded as a significant threat to organizations in terms of employee continuity. In this respect, turnover intention, which has been recently focused on in the literature, refers to the fact that an employee considers leaving their current job for various reasons and has a tendency to do so. This behavior, which is supposed to be a measure of the employee’s turnover process, is also thought to be an indicator of the employee’s satisfaction level at work, organizational commitment, and career expectations ([56]).

Even though turnover intention is often confused with turnover, it is a distinctly separate concept. Whereas workforce turnover refers to the actual departure of the employee from the workplace, the turnover intention is a thought process that is shaped by the mental processes of the individual and has the potential to turn into behavior but has not turned into action yet ([14]). In other words, the individual’s turnover intention is as follows: Should I leave my job? What are my alternatives? Should I put my decision into action? It is a psychological process that seeks answers to questions. This process reflects the evaluations made by the individual between the negativeness they feel in their current job position and the attractive opportunities in the external environment ([75]).

The turnover intention has been defined in the literature from different perspectives by different researchers. When approached from a cognitive perspective, the concept refers to the process of evaluating the possibility of leaving one’s current job and making a conscious decision in this direction ([37]). When the concept is considered from the psychological reaction dimension, the turnover intention is expressed as a reflection of the individual’s internal dissatisfaction and negative emotional reaction to organizational conditions. Another definition describes it as a critical criterion that shows the success level of organizations’ workforce planning and employee engagement strategies and directs these strategies ([34]).

When the findings of studies conducted in order to determine the reasons for turnover intention are considered, it is realized that these are generally based on individual, organizational, and environmental factors. It has been determined that an individual’s perceptions of their job such as psychological capital, job satisfaction, and career expectations affect the turnover intention directly ([32]). Organizational dynamics such as working conditions, leadership style, and justice are suggested to play a critical role in an individual’s turnover intention. Furthermore, environmental factors such as labor market conditions, economic uncertainties, and social norms are claimed to determine the context of turnover intention ([47]).

### 2.3. Quiet Quitting

Quiet quitting, a relatively new concept in the organizational behavior literature, was first put forward by Boldger at the Texas A&M Economics Symposium held in 2009 and refers to an approach that defines the reflection of employees’ low commitment to their jobs or organizations on business processes ([19]). According to the generally accepted definition, quiet quitting can be described as the employee reducing the effort they show, losing their commitment to their job, fulfilling only the basic requirements, and not tending to perform any work that is not rewarded though they actually continue their job ([33]). Quiet quitting does not express the employee’s turnover intention, but rather emotional withdrawal and loss of motivation towards the job and the organization. Employees in a quiet quitting situation tend to avoid taking too much responsibility, become insensitive to development or promotion opportunities, and develop a tendency to complete only what is adequate at work ([38]).

Quiet quitting and resignation (physical resignation), which are often confused owing to their conceptual similarity, are two very different concepts. In this respect, quiet quitting refers to the fact that the employee continues to stay at the workplace but mentally and emotionally is detached from their job. In contrast, physical resignation refers to the fact that the employee leaves their job and looks for another job. In this case, the employee fulfills their duties, although their contribution to the workplace and commitment to their job decrease. Moreover, quiet quitting should not be regarded as a sign of laziness or dissatisfaction. People with this tendency do their jobs, and sometimes even work harder than others, but due to this dedication, they may believe that their labor is being exploited. The quiet quitting that begins with this belief can emerge as a protection reflex for the employee ([71]).

When the reasons for its occurrence are taken into consideration, it can be realized that quiet quitting may stem from factors such as lack of work–life balance, low job satisfaction, burnout syndrome, inadequate recognition, and high workload. Situations such as the employee feeling that they are not valued at work or do not receive sufficient recognition or reward for their high efforts increase the tendency for quiet quitting. In terms of organizational effects, quiet quitting can have significant costs in the long term. Reducing employee effort and productivity can pave the way for decreased productivity in the workplace and weakened team dynamics. As a result, this situation can reduce motivation by causing workload asymmetry among colleagues and damage organizational identification by influencing employees who do not show this tendency, and it causes damage to the corporate culture ([78]).

## 3. The Relationship Between Concepts

### 3.1. The Relationship Between AI Anxiety and Turnover Intention

As stated before, AI anxiety refers to the uncertainty and anxiety that individuals feel about the integration of AI technologies into business life. Turnover intention represents individuals’ cognitive and emotional tendencies to leave their current jobs. It is emphasized in the literature that this trend may directly or indirectly stem from AI anxiety. It has been found that individuals who feel anxious at work for any reason have a significantly higher intention to leave their jobs. Moreover, the turnover intention has been observed to increase when individuals feel their job security is threatened. They experience emotional exhaustion and their tendency to leave the job increases, and it is suggested that this situation becomes especially more evident in sectors where AI is used in business processes extensively ([47]).

In this respect, the uncanny valley theory leads to an increase in anxiety and fear in cases where the outputs of AI are not fully understood, thus increasing the turnover intention. It is claimed that in organizations that use AI, not being informed and supported adequately by employees triggers AI anxiety, and low organizational support triggers the turnover intention ([70]). Based on the aforementioned research findings and literature readings, the following H1 hypothesis was formed.

**H1.** 
*AI anxiety has a positive significant effect on turnover intention.*


### 3.2. The Relationship Between AI Anxiety and Quiet Quitting

It is emphasized in the literature that AI anxiety and quiet quitting are related to each other and that this relationship can be explained especially within the scope of psychological and organizational factors in the working environment ([44]). It has been found that AI anxiety reduces employees’ job satisfaction and weakens their organizational commitment. Nevertheless, it is suggested that employees show a tendency to engage in quiet quitting behavior when they believe that AI threatens their jobs ([65]). It has been found that the rapid integration of AI into organizational processes and practices can bring about psychological contract violations in employees, which in turn triggers quiet quitting ([27]). It is also highlighted that the balance between job demands and available resources shapes the relationship between AI anxiety and quiet quitting. For this reason, inadequate resources and high demands can increase quiet quitting by making it difficult for employees to cope with the effects of AI. AI anxiety has been found not to influence quiet quitting directly but to trigger burnout in employees and support employees’ quiet quitting behaviors through burnout, that is, an indirect effect mechanism operates. Considering this, the following H2 hypothesis was formed based on the research findings and literature readings.

**H2.** 
*AI anxiety has a positive significant effect on quiet quitting behavior.*


### 3.3. The Relationship Between Quiet Quitting and Turnover Intention

It is clearly realized from the definitions of the concepts in question that quiet quitting and turnover intention are two concepts that are highly related to each other. Actually, quiet quitting refers to employees’ emotional distance from work by fulfilling their responsibilities at work at a minimum level whereas turnover intention refers to a cognitive tendency for individuals to consider leaving their current jobs ([22]). It is clear that both concepts refer to the process closest to actually leaving a job. However, the relationship between these two concepts has been extensively studied in the literature, especially in terms of job satisfaction, organizational commitment, and burnout. In this regard, [21] ([21]) suggested that in working environments where the level of quiet quitting is high, employees have a higher tendency to leave their jobs; therefore, there is a strong positive relationship between the concepts.

In a study conducted by [27] ([27]), the relationship between quiet quitting and turnover intention was associated with psychological contract violations and perceptions of organizational justice. In this respect, they put forward that quiet quitting behaviors emerge when employees’ psychological contracts with the organization are damaged, which triggers the turnover intention. Withdrawal Progression Theory suggests that withdrawal behavior in organizations progresses from less severe forms to more severe forms and that the result will continue until the employee leaves the job. While tardiness and absenteeism are milder withdrawal behaviors, the turnover intention is regarded as more severe behavior ([41]). Within the scope of the present study, quiet quitting is considered to be a preliminary stage for the turnover intention. In another study conducted by [18] ([18]) with university information technology employees, it was found that disengagement and burnout increased quiet quitting behaviors and that these three concepts were strongly related to the turnover intention. Based on the research findings and literature readings, the following H3 hypothesis was formed.

**H3.** Quiet quitting has a positive significant effect on turnover intention.

### 3.4. The Mediating Role of Quiet Quitting in the Relationship Between AI Anxiety and Turnover Intention

In terms of its conceptual dynamics, quiet quitting draws attention as a critical concept in understanding the relationship between AI anxiety and turnover intention. Quiet quitting causes employees to show performance at a minimum level as their commitment to the workplace decreases and manifests itself as an antecedent of the turnover intention. Studies in the literature make it possible to estimate that quiet quitting is a mechanism that increases or mediates the effect of AI anxiety on turnover intention and further suggests that the concepts in question can only be understood with a holistic approach in this direction ([27]).

When the relationship between concepts is considered from this perspective, it is realized that AI anxiety increases employees’ concerns about job security, reduces their emotional commitment to the organization, and triggers quiet quitting behavior, while quiet quitting weakens employees’ connection with the organization, reduces job satisfaction, and increases their turnover intention. For this reason, quiet quitting, as a kind of “transitional phase”, was estimated to play a role in strengthening the relationship between AI anxiety and turnover intention, and in this regard, the H4 hypothesis of this research was formed.

**H4.** 
*Quiet quitting has a mediating role in its effect on AI anxiety and turnover intention.*


The research model including the hypotheses is presented below (Figure 1).

## 4. Method

The population of this study consists of employees from small- and medium-sized enterprises operating in Kırıkkale province. According to the 2023 data of the Turkish Employment Agency, about thirty thousand people are employed in SMEs in Kırıkkale province ([29]). The formula proposed by [15] ([15]) in Formula (1) was utilized to determine the sample in question. In accordance with the literature, a confidence level of 95% and a margin of error of 5% were used to ensure statistical reliability and generalization of the findings ([57]). Moreover, it is an effective method for calculating the sample numbers of smaller groups as well as calculating large masses with the formula ([17]). The n expression in the formula expresses the population size, the p expression expresses the percentage of occurrence of a situation or condition, the e expression expresses the margin of error, and the z expression expresses the confidence level (1.96% in the 95% confidence interval) ([3]).(1)$$n_{0}=\frac{Z^{2}pq}{e^{2}}$$

According to the formula mentioned above, at least 379 people must be included. Additionally, considering that there are 42 questions in this study, including demographic variables, it is thought that the number of questions should be ten times higher according to the literature ([24]). The study was conducted in Kırıkkale because of the city’s geographical proximity to Ankara and transportation facilities that connect many cities and strategic features ([39]). SMEs are organizations of great significance for the development of trade in the world. When the literature is considered, they are realized to form the basic dynamics of the economy ([1]). According to the Turkish Statistical Institute, SMEs are defined as enterprises that employ less than two hundred and fifty people and whose balance sheet does not exceed TRY 250 million. They constitute 99.7% of all enterprises in Turkiye and contribute 36.4% to added value ([79]). SMEs are thought to have such an important place in the country’s economy that it is thought reasonable to conduct a study on these organizations.

Applications of AI by multinational organizations have long attracted attention. SMEs, on the other hand, have realized the potential of this technology with the development of technology, more affordable costs, and the provision of access to AI technologies ([61]). AI is different from other information technologies in that it is a technology that can learn, adapt, and develop. It is of great importance for SMEs due to the standardization of tasks and increasing the efficiency of the organization with its data analytics power. Chatbots, in particular, contribute to the flexibility of the company and customer satisfaction for SMEs ([84]). The development of SMEs is a fundamental approach for achieving economic growth in Turkiye. Successful adoption of technology is vital for SMEs to achieve and maintain sustainable competitive advantages ([6]). The present study was designed with the understanding that the future and development of SMEs would be influenced by AI technologies.

The present study utilized a socio-demographic information form, the AI Anxiety Scale, Turnover Intention Scale, and Quiet Quitting Scale as data collection tools. A 5-point Likert-Type Scale ([46]) was used for data collection (1 = Strongly Disagree to 5 = Strongly Agree). The AI Anxiety Scale was developed by [83] ([83]) and adapted into Turkish by [77] ([77]). The scale consists of twenty-one statements and four dimensions. The Cronbach Alpha value of the original scale was determined to be 0.960. The scale includes statements such as “I am worried about learning to understand all the special functions associated with an AI technique/product”, and “I find humanoid AI techniques/products (e.g., humanoid robots) threatening”.

The Turnover Intention Scale was developed by [53] ([53]) and was adapted into Turkish by [62] ([62]). The scale consists of three statements. The Cronbach Alpha value of the original scale was determined to be 0.904. In the scale, there are statements such as “I often think of leaving my current job” and “I am actively looking for a job in other institutions”. The Quiet Quitting Scale was developed by [5] ([5]) and consists of thirteen statements. The Cronbach Alpha value of the original scale was determined to be 0.810. In the scale, there are statements such as “Even though I do work other than what is shown in my job description and contribute to my workplace, I cannot get enough compensation” and “Even though overtime is constantly applied in the institution I work for, overtime is not paid”.

The scales were collected digitally and face-to-face from SMEs in Kırıkkale between 25 December 2024 and 15 January 2025. Convenience sampling was preferred to select 457 people working in 21 SMEs in the service sector and 48 SMEs in the manufacturing sector. Convenience sampling provides easy access to the participants in the sample and ensures effective data collection ([58]). As the convenience sampling is based on voluntary participation, it is a method that is likely to include people with strong knowledge and feelings about the subject matter ([73]). The fact that the study data were obtained from SMEs located in only one province in Turkiye with a cross-sectional understanding constitutes an important limitation in terms of generalizing the findings. The data obtained after the survey were analyzed through Smart-PLS version 3.2.9. This application was preferred due to its suitability for evaluating models with a large number of components and indicators ([57]). In this regard, Smart-PLS is a frequently preferred method owing to the simultaneous estimation of multiple and interrelated dependent relationships between variables and the measurement of latent structures at the same time ([66]). Before the survey application, permission was obtained from the Yozgat Bozok University Ethics Committee on 19 December 2024 with the letter numbered 20/23.

## 5. Findings

For the present study, 457 people were included, and detailed socio-demographic information about the individuals participating in the study is presented in Table 1. More than half of the participants (53.8%) were women and the rest (46.2%) were men. In general, the prevalence of female employment stands out compared to the conditions in Turkiye, where male employment is higher. At this point, it can be said that increasing the number of women entrepreneurs with the grants and supports of the Small and Medium Enterprises Development and Support Administration (KOSGEB), as well as highlighting positive discrimination towards women’s employment, is effective ([40]). It is seen that the number of single participants (54.0%) is higher than the number of married participants. It was found that most of the participants in this study (33.5%) were high school graduates and almost half of them, 49.9%, had 5 years or less of experience. Furthermore, it is observed that the majority of the participants (72.2%) do not have an administrative position.

The validity of the scales was analyzed through the Smart-PLS program, and the relevant acceptable values are presented in Table 2. Factor loadings are observed to be greater than 0.50; Cronbach’s Alpha, Composite Reliability, and rho_A coefficients are found to be greater than 0.70. The AVE value is required to be higher than 0.50 ([25]). Internal consistency analyses have been carried out in order to evaluate the consistency of the underlying structure of the indicator elements in a research model. When the literature is considered, Cronbach’s Alpha coefficient and the Composite Reliability coefficient, expressed as CR, emerge as reliability coefficients. However, it is also recommended to calculate the rho_A coefficient in studies conducted with PLS-SEM. When factor loading values are considered, it is realized that the generally accepted value in the literature is above 0.50, which results from the fact that it should consist of strong items with factor loadings higher than 0.50 ([68]). The fact that the validity and reliability values in Table 2 are above the threshold values for all constructs indicates the high reliability of the internal consistency and convergent validity values of the scales ([57]).

As the variance inflation factor (VIF) value presented in Table 2 was below 10, it was found that there was no common method bias ([59]). By evaluating the obtained results in the light of threshold values, it is realized that the scales do not have multicollinearity and common method bias. For this reason, there is no need to perform a confirmatory factor analysis such as CB-SEM in the Smart-PLS program. When the relevant model is formed and the necessary steps are followed, these values are automatically shown by the program to the researcher, which confirms that the program is a user-friendly one ([25]). As a result of the analysis, statements 10 and 13 from the Quiet Quitting Scale and statements 1, 2, and 7 from the AI Anxiety Scale were excluded from the analysis as they did not meet the factor loading value of 0.5.

Discriminant validity is a criterion used to determine the extent to which latent variables are distinguished from other latent variables. When the literature is examined, the criteria proposed by [20] ([20]) and [28] ([28]) are known as the most frequently used discriminant validity method. Discriminant validity values according to Fornell and Larcker and Heterotrait–Monotrait criteria are illustrated in Table 3.

In the Fornell–Larcker Criterion, the coefficients at the intersection of the variables express the square root of the AVE coefficients of the relevant variables, and discriminant validity is realized to be ensured. The test is important in that it evaluates whether there is a high overlap between the measurement variables of a model and whether the distinction between the structures is sufficient, and it is observed that the distinction is achieved ([58]). On the other hand, HTMT, which has been recently proposed in PLS-SEM research, suggests that the average correlations between research structures are indicated by calculating the geometric mean of the average correlations within the expressions of the structures. According to the Heterotrait–Monotrait Ratio (HTMT) criterion, the threshold value should be below 0.90 ([28]) and as the values are below 0.90 as presented in Table 3, discriminant validity has been achieved. In the study, all variables were significantly below the specified threshold in terms of HTMT ratio values, indicating that each scale structure is different and separate ([57]).

Following the evaluation of the measurement part of the research model, the “Bootstrapping” calculation method and a bootstrapping resampling number of 5.000 in the Smart-PLS program were preferred in the structural model evaluation. The bootstrap test developed by [16] ([16]) was utilized to perform the mediation test. The robust bootstrapping method helps in terms of predicting standard errors and confidence intervals for path coefficients accurately. At the same time, the method provides a reliable basis for hypothesis testing ([57]). As a result of the test, beta, *p*, and t values were examined so as to test whether the path coefficients were statistically significant ([25]). Figure 2 presents the Smart-PLS diagram obtained from the research model.

The structural model is displayed in Figure 2, and when the goodness-of-fit values are examined, the value of SRMR < 0.080 is 0.064, the d_ULS value is 2.149, and the d_G value is determined to be 0.801. The Chi-Square < 5 value was found to be 2.111 and the NFI > 0.80 value was found to be 0.808. In this respect, the goodness-of-fit values obtained from the model are observed to be within acceptable limits. Regarding the relevant values, it is suggested that the Standardized Root Mean Square Residual (SRMR) value should be below 0.08 and the Normed Fit Index (NFI) value should be above 0.80 ([10]). Structural equation model coefficients are illustrated in Table 4.

Table 4 presents the coefficients that indicate the mediating role of quiet quitting in the effect of AI anxiety on turnover intention. Before the mediating variable was added to the model, AI anxiety had a positive and significant effect on turnover intention (ß = 0.390 *p* < 0.01) and these values reveal the total effect. With the addition of quiet quitting into the first model, it was observed that the effect of AI anxiety on turnover intention was not significant (ß = 0.068; *p* > 0.05). It was also realized that there is a positive and significant effect of AI anxiety on quiet quitting (ß = 0.485; *p* < 0.01) while quiet quitting has a positive and significant effect on turnover intention (ß = 0.6563; *p* < 0.01). The significance of the addition of the mediating variable into the model and the decrease in beta values was determined through the VAF test. The VAF value takes values between 20% and 80% (0–20% no mediation; 20–80% partial mediation; 80–100% full mediation).

VAF is calculated by dividing the indirect effect by the total effect plus the indirect effect, and when calculated according to the formula, the VAF value is found to be 82%. For this reason, it can be inferred that quiet quitting has a full mediating role in the effect of AI anxiety on turnover intention. Moreover, R^2^ and Q^2^ values obtained within the scope of the model are shown in Table 4. The relevant values indicate the predictive powers of the scales used in the model. The R^2^ value results of quiet quitting show that AI anxiety explains 23.5% of the variation in quiet quitting. The results of the R^2^ value of turnover intention indicate that AI anxiety and quiet quitting explain 48.8% of the variation. The R^2^ value is stated to be evaluated through interdisciplinary approaches, and the Q^2^ value should be above zero ([25]). The Q^2^ value indicates a model’s ability to predict future or out-of-sample data accurately ([57]). It can be realized that the predictive power values presented in Table 4 are compatible with the literature. Effect size (f^2^) refers to a measure of the strength of the relationship between variables. The expression f^2^ is a measure of a kind of standardized mean effect in the population across all levels of the independent variable ([13]). A value of f^2^ greater than or equal to 0.02 indicates a low level, a value greater than or equal to 0.15 indicates a moderate level, and a value greater than and equal to 0.35 indicates a high level of association ([26]). It was observed that there was a moderate level (0.307) between AI anxiety and quiet quitting, a low level (0.070) between AI anxiety and turnover intention, and a high level (0.658) between quiet quitting and turnover intention.

## 6. Discussion

The findings from the present study make a great contribution to the literature. When the first hypothesis was considered, it was realized that AI anxiety did not have a positive and significant effect on turnover intention. The significant effect of AI anxiety on turnover intention at ß = 0.390 became insignificant, decreasing to ß = 0.068 through quiet quitting. The fact that AI makes autonomous decisions, works without human intervention, and disregards some ethical sensitivities makes employees anxious. Long-term analysis of the working outputs produced by employees with AI is also regarded as one of the disturbing actions ([25]). Furthermore, applications such as personalized virtual chat robots and virtual customer services have taken their place in many sectors, especially in banks, and this brings concerns about losing jobs ([82]). It is predicted that job loss will affect 47% of employees in America in the coming period due to AI. Considering that 360,000 to 670,000 people lose their jobs annually as the cost of robots decreases, AI is expected to have a more devastating effect ([35]). It is observed that AI increases the turnover intention due to anxiety, fear, and psychological interaction that emerge as a result of the effects in real life. Even though the anxiety that arises for various reasons increases the intention of employees to leave their job, it is seen that the finding obtained is reasonable when the fact that the main reason behind their AI anxiety is concern about losing their job is considered.

In accordance with the second hypothesis, it was found that AI anxiety had a positive and significant effect on quiet quitting. The fact that AI anxiety has a positive and significant effect on quiet quitting reveals that employees experience uncertainty about their job security and professional roles due to AI integration. This anxiety can reduce the motivation of employees and weaken their organizational commitment. Instead of leaving the job outright, employees can start reducing their commitment to their work with only minimal effort and avoiding additional responsibilities ([64]). Employees can react emotionally or physically to the changes in the organization, and some reactions should be considered to occur with the inclusion of AI in the organization. The AI chatbot ChatGPT reached 100 million users in just two months and influenced many economic, cultural, and philosophical elements ([67]). Bearing this in mind, the result obtained can be possibly attributed to the changes experienced in some sectors as a result of the widespread use of AI and employee concerns. When the control of the outputs produced by AI, its ethical dimensions, and its human effects are considered, it seems likely that the uncertainty experienced in the reality and inconsistency of AI will lead to an uncanny valley effect on people ([49]). When the literature is considered, it is realized that there are many studies supporting this result. As a matter of fact, in the study conducted by [44] ([44]), it was observed that when employees see the impact of technology on their work as a threat, they develop a tendency toward quiet quitting. In a similar study carried out by [76] ([76]), it was suggested that the negative thinking developed about AI in employees leads to emotional exhaustion, reduces their organizational commitment, and accelerates the turnover process. In order to overcome this situation, open communication and transparent management processes should ensure that employees see AI as a support tool, not as a threat. Furthermore, it can be recommended to organize in-service training programs related to AI, to ensure a gradual transition to AI technologies, not sharp, and to develop support strategies for employees.

Considering the third hypothesis, quiet quitting was found to have a positive and significant effect on turnover intention. When the literature was examined, the effect experienced was associated with the Withdrawal Progression Theory, and it was observed that quiet quitting was an early withdrawal behavior and turned into more serious forms of withdrawal, leading to the turnover intention ([36]). Quiet quitting behavior is a concept that weakens employees’ emotional commitment to the organization over time and increases their likelihood of considering leaving the organization completely. In the study carried out by [21] ([21]) on nurses, it was observed that 49.92% of the employees who had the intention to leave their job also reported quiet quitting, while only 26.8% of the employees who intended to leave the job did not experience quiet quitting. In the study conducted by [23] ([23]), a medium-level relationship was detected between quiet quitting and turnover intention with a ratio of ß = 0.313. The findings obtained in the third hypothesis of this study are consistent with the literature. This finding reveals the importance of SMEs establishing effective feedback mechanisms to identify employees’ quiet quitting tendencies. Actually, the sooner the factors that lead to quiet quitting are recognized, the more effective the steps to prevent employees from leaving the job can be. In addition, as a precaution against quiet quitting, it is recommended that SME managers provide meaningful career development opportunities to their employees, develop a culture of appreciation, and create strategies to increase job satisfaction. In accordance with the fourth hypothesis, it was found that quiet quitting had a mediating role in the effect of AI anxiety on turnover intention. The pressure, stress, and unhappy events experienced by employees in the workplace trigger a negative emotional reaction. Therefore, the experiences motivate employees to withdraw and disconnect from their jobs in order to preserve their psychological and emotional resources ([85]). As AI is regarded as uncanny and uncertain, it is realized that employees first turn their attention to quiet quitting as a manifestation of the anxiety they experience within themselves, and in the following process, the turnover intention is felt more.

## 7. Conclusions

The present study aims to determine the effect of artificial intelligence anxiety on turnover intention and to examine the mediating role of quiet quitting in this relationship, through a survey carried out with 457 employees in SMEs located in Kırıkkale, Turkiye. The data obtained within the scope of the study were analyzed through structural equation modeling using the Smart-PLS program, and the findings made significant contributions to the existing literature.

Based on the model created by the study, it was found that artificial intelligence anxiety did not have a significant direct effect on the turnover intention, but it had an indirect effect through quiet quitting. Based on this, the psychological effects of artificial intelligence-induced uncertainties on employees and the concerns brought about by uncertainty result in a decrease in work motivation and a weakening of organizational commitment. In particular, the increase in the perception of job insecurity reduces employees’ interest in their jobs and triggers a tendency toward quiet quitting. This finding reveals that artificial intelligence anxiety is a critical factor that strengthens employees’ turnover intention in the long term and overlaps significantly with the previous literature ([21]).

Moreover, this study indicated that quiet quitting had a positive and significant effect on turnover intention. Quiet quitting causes individuals to become emotionally detached from their organizations and have a stronger tendency to leave over time. This reveals that supportive measures provided in the workplace and confidence-increasing policies for employees are critical. In this regard, research shows that quiet quitting is linked to lower motivation at work, increased stress levels and lower employee satisfaction. At the same time, it is realized that the significant effect of AI anxiety directly on turnover intention becomes insignificant with the full mediation effect of quiet quitting. In this respect, quiet quitting resignation seems to be an intermediate stage before the turnover intention and has a high impact on the employee’s turnover intention, which is consistent with previous studies in the literature ([27]).

The study also shows that artificial intelligence anxiety has a positive and significant effect on quiet quitting. This finding suggests that when employees perceive artificial intelligence as a threat to job security, they reduce their commitment to their jobs. In particular, low organizational support and uncertainty about career opportunities further reinforce this effect. Similarly, in the literature, employees’ perceptions of job insecurity and career uncertainty are considered to be significant factors that increase the tendency toward quiet quitting ([65]).

## 8. Managerial/Practical Implications

The findings of this study provide basic insights for SME managers and practitioners in Turkiye by emphasizing the practical applications required to make strategic decisions regarding human resources, which are considered the most critical resource of organizations, and to manage the organizational adaptation and commitment processes of employees effectively. Some of the managerial approaches recommended to minimize employee concerns and reduce turnover intention during the transformation process of artificial intelligence and business processes can be listed as follows.

First of all, a human-centered artificial intelligence strategy should be developed by the management. In order to address employees’ concerns during the integration of artificial intelligence into the workforce, managers should enable open communication channels. Awareness programs should be organized for employees, and it should be explained that artificial intelligence is not a threat but a tool that supports business processes. In this regard, training programs that will increase the knowledge level of employees in the technological transformation process should be encouraged.

Moreover, managers should develop interaction mechanisms to understand their concerns by communicating with employees one-on-one. A management approach based on empathy will play a critical role in preventing quiet quitting by strengthening employee loyalty. In addition, it is also of great importance to present career planning and development opportunities transparently. Managers should practice transparent promotion policies and personalized career development plans to increase employees’ long-term motivation.

Furthermore, performance-based incentives, social recognition practices and making employee contributions visible will increase employees’ motivation and strengthen their commitment to the organization. On the other hand, ensuring the psychological safety of employees is another significant factor in terms of preventing quiet quitting. Adopting open communication policies and creating a safe working environment where employees can express their concerns freely will strengthen employee loyalty. Mechanisms should be created where employees can receive psychological support, especially so as to manage the uncertainties brought by artificial intelligence integration.

Including employees in decision-making processes is another important step that increases organizational commitment and is also effective in decreasing perceived uncertainty. Active participation of employee representatives in management processes and management strategies based on employee feedback will strengthen employees’ belonging to the organization and pave the way for a decrease in the turnover rate. In this respect, a democratic approach should be adopted in management processes and employees’ contributions should be encouraged.

These strategic approaches will help managers reduce turnover intention and prevent quiet quitting by increasing employee satisfaction. In the long term, the trust environment and employee-oriented management approach provided within the organization will have direct positive effects on the sustainable growth and success of businesses.

## 9. Theoretical Implications

With the help of the present study, the effects of AI anxiety on quiet quitting and turnover intention were tried to be explained through the uncanny valley theory. Although the uncanny valley theory is mostly associated with autonomous robots in the literature, the study has revealed that AI also supports the relevant theory. The uncanny valley theory suggests that humans mostly initially have a positive reaction to robots or AI elements that perform tasks attributed to humans, but when this similarity crosses a certain threshold, they feel uncomfortable and anxious. This theory can be directly related to AI anxiety as employees may develop a threat perception subconsciously because they see AI technologies approaching human competencies. In this respect, AI anxiety is shaped by concerns such as employees’ fear of losing their jobs, technology becoming uncontrollable, or human skills being devalued ([69]). Based on the theory, it is realized that 0.322 of the effect between AI anxiety and turnover intention is explained by quiet quitting. It enhances our understanding of how workplace policies affect employees’ turnover intention, and highlights the importance of assessing quiet quitting as a precursor to turnover intention. In this regard, one more study including an application has been added to the few studies that associate AI applications with the uncanny valley theory ([52]; [74]). In the study, the effect of quiet quitting on turnover intention was explained through the Withdrawal Progression Theory, which suggests that employees withdraw as a result of the problems they experience in the organization. In this respect, it is suggested that as the problems accumulate and increase, a quiet quitting process emerges, and as the problems continue, a more severe withdrawal and turnover intention occurs ([36]).

## 10. Limitations and Directions for Future Research

It has been proven by this study that if employees are not motivated enough in cases of quiet quitting, it will lead to emotional and physical depression and make employees concentrate on their turnover intention. Within the scope of the study, it was also determined that quiet quitting is a preliminary stage for the intention to leave the job.

Conducting the study with a cross-sectional approach on SMEs located in only one province in Turkiye constitutes a significant limitation. It is accepted that the participants in the study had limited comprehension ability during the survey, and there are some limitations in terms of generalizing the results.

In line with the findings of this study, several important suggestions can be made for future studies. First, it is recommended to conduct comparative studies in different sectors and business lines in order to understand the impact of AI anxiety on quiet quitting and turnover intention. By examining the differences between the anxiety levels of employees in technology-intensive sectors and the concerns of employees in more traditional business lines, unique strategies can be developed on a sectoral basis. Secondly, in this study, the mediating role of quiet quitting between AI anxiety and turnover intention was determined. However, variables such as organizational factors (leadership style, working conditions, perception of job security) and individual factors (resilience, motivation to learn, technological adaptability) that may affect this process should be addressed in future research. Third and finally, as the current study is based on quantitative methods, it is recommended to use qualitative research methods (e.g., in-depth interviews and case studies) to gain a deeper understanding of how AI anxiety shapes employees’ emotional and cognitive processes. Thus, it will be possible to evaluate the experiences of employees and how their concerns are reflected in their daily work practices more comprehensively.

## Figures and Tables

**Figure 1 behavsci-15-00249-f001:**
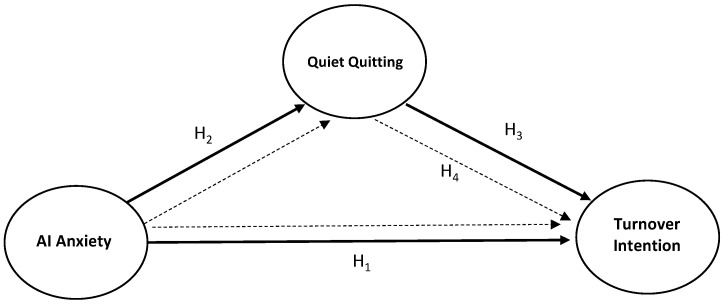
Research model (source: developed by the authors).

**Figure 2 behavsci-15-00249-f002:**
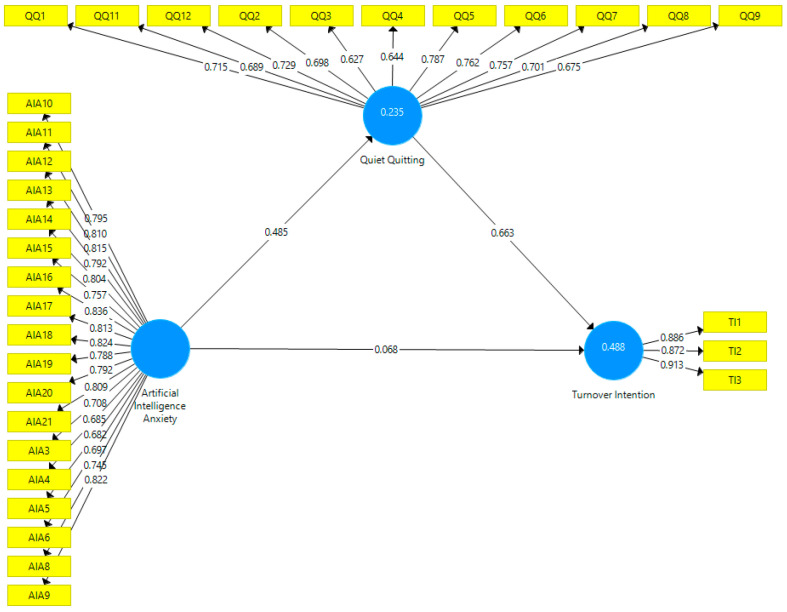
Path diagram (source: developed by the authors).

**Table 1 behavsci-15-00249-t001:** Socio-demographic information.

Items		*n*	%
Gender	Female	246	53.8
	Male	211	46.2
Age	Between 18 and 24 Years of Age	136	29.8
	Between 25 and 29 Years of Age	106	23.2
	Between 30 and 34 Years of Age	70	15.3
	Between 35 and 39 Years of Age	59	12.9
	40 Years and Above	86	18.8
Marital Status	Married	210	46.0
	Single	247	54.0
Level of Education	Elementary	43	9.4
	High School	153	33.5
	Undergraduate	115	25.2
	Bachelor	113	24.7
	Master	25	5.5
	Doctorate	8	1.8
Years of Experience in the Organization	Between 1 and 5 Years	228	49.9
	Between 6 and 10 Years	93	20.4
	Between 11 and 15 Years	56	12.3
	Between 16 and 20 Years	40	8.8
	21 Years and Above	40	8.8
Administrative Position in the Organization	Yes	127	27.8
	No	330	72.2

**Table 2 behavsci-15-00249-t002:** Factor loading values, reliability, and validity.

Variable	Factor Loadings	Mean	Standard Deviation	Kurtosis	Skewness	VIF
Quiet Quitting Cronbach’s Alpha = 0.901, rho_A = 0.906, CR = 0.917, AVE = 0.503						
QQ1	0.715	3.013	1.326	−1.242	−0.058	1.841
QQ2	0.698	2.753	1.312	−1.169	0.283	1.745
QQ3	0.627	2.961	1.299	−1.172	0.037	1.584
QQ4	0.644	2.814	1.249	−1.067	0.214	1.658
QQ5	0.787	2.893	1.291	−1.186	0.096	2.156
QQ6	0.762	2.694	1.327	−1.109	0.339	2.025
QQ7	0.757	3.070	1.359	−1.305	0.009	2.074
QQ8	0.701	3.204	1.313	−1.137	−0.287	1.791
QQ9	0.675	3.260	1.315	−1.044	−0.356	1.814
QQ11	0.689	3.376	1.311	−0.981	−0.445	1.789
QQ12	0.729	2.989	1.270	−1.130	−0.082	1.800
Turnover Intention Cronbach’s Alpha = 0.869, rho_A = 0.873, CR = 0.920, AVE = 0.793						
TI1	0.887	2.705	1.267	−1.027	0.297	2.178
TI2	0.870	2.571	1.238	−0.873	0.462	2.203
TI3	0.914	2.525	1.236	−0.935	0.368	2.709
AI Anxiety Cronbach’s Alpha = 0.961, rho_A = 0.962, CR = 0.965, AVE = 0.605						
AIA3	0.708	2.567	1.209	−0.863	0.394	2.999
AIA4	0.685	2.578	1.190	−0.821	0.393	3.118
AIA5	0.682	2.575	1.190	−0.813	0.374	2.854
AIA6	0.697	2.516	1.187	−0.738	0.461	2.966
AIA8	0.745	2.827	1.282	−1.092	0.195	2.240
AIA9	0.822	3.013	1.239	−1.008	−0.080	3.333
AIA10	0.795	3.092	1.293	−1.085	−0.099	3.053
AIA11	0.810	3.079	1.287	−1.053	−0.055	3.325
AIA12	0.815	3.090	1.297	−1.079	−0.131	3.588
AIA13	0.792	2.921	1.289	−1.072	0.092	2.634
AIA14	0.804	3.083	1.297	−1.088	−0.083	3.137
AIA15	0.757	3.265	1.355	−1.148	−0.266	2.734
AIA16	0.836	3.055	1.276	−1.056	−0.052	3.481
AIA17	0.813	3.094	1.306	−1.120	−0.098	3.221
AIA18	0.824	3.020	1.310	−1.145	0.005	3.652
AIA19	0.788	2.996	1.308	−1.132	0.038	3.192
AIA20	0.792	2.989	1.290	−1.098	0.100	3.030
AIA21	0.809	2.939	1.288	−1.097	0.078	3.392

QQ = quiet quitting; TI = turnover intention; AIA = AI anxiety.

**Table 3 behavsci-15-00249-t003:** Fornell–Larcker Criterion and Heterotrait–Monotrait Ratio (HTMT) values.

Fornell-Larcker Criterion				Heterotrait-Monotrait Ratio (HTMT)			
	1	2	3		1	2	3
QQ	0.709			QQ			
AIA	0.485	0.778		AIA	0.519		
TI	0.696	0.390	0.890	TI	0.774	0.419	

QQ = quiet quitting; TI = turnover intention; AIA = AI anxiety.

**Table 4 behavsci-15-00249-t004:** Hypothesis test results.

Paths	Estimate	Standard Deviation	*t* Values	*p*	Hypothesis
AIA → TI	0.068	0.047	1.444	0.149	H1 Reject
AIA → QQ	0.485	0.048	9.999	0.000	H2 Accept
QQ → TI	0.663	0.038	17.426	0.000	H3 Accept
AIA → QQ → TI (Indirect Effect)	0.322	0.039	8.170	0.000	H4 Accept
AIA → TI (Total Effect)	0.390	0.045	8.669	0.000	
R^2^ = (QQ = 0.235; TI = 0.488)					
Q^2^ = (QQ = 0.117; TI = 0.381)					

QQ = quiet quitting; TI = turnover intention; AIA = AI anxiety.

## Data Availability

The data that support the findings of this study are available from the corresponding author upon reasonable request.

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
