# Peer review of "Assessing the Effect of Artificial Intelligence Anxiety on Turnover Intention: The Mediating Role of Quiet Quitting in Turkish Small and Medium Enterprises"

_behavsci, 2025, doi:10.3390/bs15030249_

Round 1
Reviewer 1 Report
Comments and Suggestions for Authors
Overall, I believe the article provides a timely and relevant empirical and theoretical contributions to the literature. The study makes a valuable contribution to the understanding of AI's psychological effects in the workplace. However, its findings should be interpreted with caution due to methodological limitations. Of course, there is always room for improvement.
Future research should track employees over time to better establish the causal relationship between AI anxiety, quiet quitting, and turnover intention. It could also attempt to include different industries and sector to enhance external validity. Moreover, while the study attributes quiet quitting to AI anxiety, other workplace factors (e.g., poor leadership, low compensation, lack of career development) might also contribute to the research. A broader model including these factors would have strengthened the analysis​. Finally, this study briefly mentions ethical concerns related to AI (e.g., job displacement, algorithmic bias) but does not deeply engage with these issues, which are critical in understanding AI anxiety​.
Author Response
Dear Reviewer 1
First of all, I would like to express my gratitude for your precious comments and suggestions to improve the article. I took your suggestions into consideration and made the following changes in the article;
Comments 1: Overall, I believe the article provides a timely and relevant empirical and theoretical contributions to the literature. The study makes a valuable contribution to the understanding of AI's psychological effects in the workplace. However, its findings should be interpreted with caution due to methodological limitations. Of course, there is always room for improvement.
Response 1: We agree with this comment. This change can be found Page 14, Line 548-551; Page 15, Line 568-587 and Page 16, Line 606-628. Changes are indicated in red font.
Comments 2: Future research should track employees over time to better establish the causal relationship between AI anxiety, quiet quitting, and turnover intention. It could also attempt to include different industries and sector to enhance external validity. Moreover, while the study attributes quiet quitting to AI anxiety, other workplace factors (e.g., poor leadership, low compensation, lack of career development) might also contribute to the research. A broader model including these factors would have strengthened the analysis. Finally, this study briefly mentions ethical concerns related to AI (e.g., job displacement, algorithmic bias) but does not deeply engage with these issues, which are critical in understanding AI anxiety.
Response 2: We agree with this comment. This change can be found Page 16, Line 629-653. Changes are indicated in red font.
Best Regards
Reviewer 2 Report
Comments and Suggestions for Authors
I appreciate the opportunity to evaluate this work and congratulate your contribution to the literature. Prior to the manuscript's consideration for publication, I recommend that the authors rectify all the criticisms raised to enhance its quality and clarity.
Title: The Mediating Role of Quiet Quitting in the Effect of Artificial Intelligence Anxiety on the Turnover Intention
Comments
The authors could consider this title: Assessing the effect of Artificial Intelligence Anxiety on Turnover Intention: The mediating role of Quiet Quitting in Turkish SMEs.
Abstract
The abstract is well-crafted; nevertheless, the authors have to incorporate a conclusion to encapsulate the principal findings and their implications.
Introduction
The introduction is well-written; however, the authors should elaborate on their contribution to the organizational behaviour literature to clearly highlight the study’s novelty and significance.
Literature review
a. The authors provide a generally accepted definition of quiet quitting but fail to acknowledge the source. Proper citation is needed to attribute the definition accurately.
b. 2.2 The authors should delete the article ‘The’
c. The authors stated that ‘It is emphasized in the literature that artificial intelligence anxiety and quiet quitting are related to each other and that this relationship can be explained especially within the scope of psychological and organizational factors in the working environment’. Authors must cite a source as they are asserting a claim grounded in current literature. Indicating that a relationship has been "emphasized in the literature" suggests that prior research has substantiated this association. To uphold academic integrity and credibility, it is imperative to reference pertinent sources that substantiate the assertion. This enables readers to validate the content and guarantees appropriate recognition of previous research.
d. The authors must furnish sufficient references to substantiate their assertions regarding the relationship between ‘Artificial Intelligence Anxiety and Quiet Quitting’ to ensure appropriate attribution and enhance the credibility of their argument.
e. In line 257, the authors reference Hamouche et al. (2023)…'; nevertheless, in line 259, they emphasize the same source as 'he puts forward…,' resulting in inconsistency. The writers must specify if 'Baa et al. (2023)' is a single-author or multi-author source and maintain consistency in citation.
f. The authors should clearly articulate the specific contributions of uncanny valley theory to this study and how it helps to understand the mediating role of quiet quitting.
g. The authors may review theses papers and cite them 1. https://www.mdpi.com/2071-1050/16/22/9980 2. https://link.springer.com/article/10.1186/s40359-024-01891-7 3. https://www.mdpi.com/2227-9032/12/3/291 to enhance the turnover concept
h. After the first introduction, use the acronym consistently throughout the manuscript. Avoid switching between the full term and the acronym, which can confuse the reader
i. The authors must ensure accurate in-text citations to uphold consistency and precision in referencing sources e.g (‘Y.-Y. Wang and Wang, 2022’) should be Wang and Wang, 2022. Authors should ensure same for subsequent ones.
j. I was expecting the authors to begin their literature review by discussing the relationship between the main independent and dependent variables first, followed by the link between the independent variable and the mediating variable in that orther. Restructuring the literature review in this order will require renumbering the hypotheses, which should also be reflected in the research model.
k. The authors should provide the source of their model eg (Authors construct, 2025).
Methodology
a. The authors failed to address the research design. It is essential for them to furnish a lucid clarification of the research strategy to enhance the understanding of the study's structure and data collection methods.
b. The authors assert that their target population is roughly thirty thousand, although they fail to cite a source for this data. For their data to be more credible, they should provide a suitable reference or describe how this estimate was arrived at. Hair et al.'s (2014) guidelines indicate that the optimal sample size should be determined by aspects such as research design, variable count, and analytical methodologies, rather than solely by population size.
c. The authors effectively cited the origins of their instruments and presented their alpha values. Nonetheless, they did not provide a rationale for the use of these instruments within the context of Turkish SMEs. The authors should elucidate the relevance and validity of these measurements for this particular context.
d. The authors failed to indicate the specific scale used to measure these items (e.g., strongly disagree to strongly agree). It is important for them to clarify the scale used to ensure transparency in their measurement approach.
e. The authors failed to address the criteria for participant inclusion and exclusion. It is imperative for them to delineate these criteria to augment the transparency and quality of their sample selection method.
f. The authors did not specify the sample method, but they mentioned distributing questionnaires to 457 individuals. The writers must delineate the sampling technique employed to elucidate the participant selection process and support the sample's validity.
g. The authors did not address the statistical software and tools employed for analysis. It is essential for them to delineate the software and analytical techniques utilized to guarantee transparency and repeatability of the study.
Findings
a. I observed that the authors utilized the SMARTPLS tool; however, this did not come up in the methodology or data analysis sections. To promote clarity and transparency, the authors should explicitly articulate their rationale for selecting SMARTPLS and detail its application. In the absence of this, it is challenging to thoroughly evaluate the validity of the results.
b. The assessment of discriminant validity was briefly mentioned, although the outcomes of the HTMT and Fornell-Larcker tests were not elaborated upon. The authors have to provide a more comprehensive explanation of these data to enhance transparency and bolster the validity of their conclusions. The authors should follow the approach outlined in these papers 1. https://www.mdpi.com/2071-1050/16/22/9980 2. https://dergipark.org.tr/tr/pub/ulasbid/article/1401710 to enhance the structure of their findings and may cite them where appropriate."
c. https://www.emerald.com/insight/2046-9012.htm 2. https://www.mdpi.com/2071-1050/16/22/9980 3. https://www.emerald.com/insight/1463-5771.htm 4. https://dergipark.org.tr/tr/pub/ulasbid/article/1401710 to enhance the structure of their findings and may cite them where appropriate."
d. The manuscript fails to address the mitigation of multicollinearity and common method bias. The authors must discuss how these issues were addressed to guarantee the robustness and validity of their findings.
e. The authors presented the R2 and Q2 values but ignored to address their implications. The authors must interpret these values within the framework of their model to explain the explanatory power and predictive significance of their findings.
f. The authors have to present and analyze the effect size (f2) values to elucidate the practical significance of their findings and the robustness of the relationships within the model.
g. The hypothesis was tested using a bootstrapping resampling of 5,000, not 5.000. The authors must amend this statement for precision.
h. Simultaneously presenting results for both direct and indirect effects, especially in mediation models, is the conventional method, as it offers a more cohesive and thorough comprehension of the relationships under examination. Nevertheless, the authors did not utilize SPSS to perform a Sobel test for this purpose. This technique is inapplicable due to the utilization of SMARTPLS. The authors should explore utilizing the bootstrapping method, commonly implemented in SMARTPLS for assessing mediation effects.
i. The authors should refrain from reporting the overall effect outcome. Given the emphasis on direct and indirect effects in mediation analysis, it is advisable to provide solely the relevant results.
j. The authors must specify the hypothesis number to correspond with the recommendations presented in the literature review (i). The hypothesis number should be incorporated into Table 4 for enhanced clarity and consistency.
Discussions
a. The authors should provide separate sections for their discussions, conclusions, theoretical implications, managerial implications, and limitations, along with directions for future research. This would help organize the manuscript more clearly and improve the flow of the content.
b. In the discussion section, the authors should explain the rationale behind their findings, assess whether their results corroborate the theory presented in the text, and delineate practical implications, especially regarding the daily management of SMEs, for each finding. While the authors examined whether their findings corroborate or contradict existing literature, a deeper consideration of the theoretical coherence and practical significance of each conclusion would enhance the discussion.
c. The authors acknowledged that the utilization of cross-sectional data was a restriction in their analysis; however, this was not addressed in the methodology section. The authors should acknowledge this restriction in the methodology section to enhance clarity regarding the study design and its possible influence on the findings.
Author Response
Dear Reviewer 2
First of all, I would like to express my gratitude for your precious comments and suggestions to improve the article. I took your suggestions into consideration and made the following changes in the article;
Comments 1: The authors could consider this title: Assessing the effect of Artificial Intelligence Anxiety on Turnover Intention: The mediating role of Quiet Quitting in Turkish SMEs.
Response 1: We agree with this comment. This change can be found Page 1, Line 2-4. Changes are indicated in red font.
Comments 2: The abstract is well-crafted; nevertheless, the authors have to incorporate a conclusion to encapsulate the principal findings and their implications.
Response 2: We agree with this comment. This change can be found Page 1, Line 35-39. Changes are indicated in red font.
Comments 3: The introduction is well-written; however, the authors should elaborate on their contribution to the organizational behaviour literature to clearly highlight the study’s novelty and significance.
Response 3: We agree with this comment. This change can be found Pages 2-3, Line 81-103. Changes are indicated in red font.
Comments 4: The authors provide a generally accepted definition of quiet quitting but fail to acknowledge the source. Proper citation is needed to attribute the definition accurately.
Response 4: We agree with this comment. This change can be found Page 5, Line 221. Changes are indicated in red font.
Comments 5: 2.2 The authors should delete the article ‘The’.
Response 5: We agree with this comment. This change can be found Page 4, Line 174. The relevant word has been deleted from the article.
Comments 6: The authors stated that ‘It is emphasized in the literature that artificial intelligence anxiety and quiet quitting are related to each other and that this relationship can be explained especially within the scope of psychological and organizational factors in the working environment’. Authors must cite a source as they are asserting a claim grounded in current literature. Indicating that a relationship has been "emphasized in the literature" suggests that prior research has substantiated this association. To uphold academic integrity and credibility, it is imperative to reference pertinent sources that substantiate the assertion. This enables readers to validate the content and guarantees appropriate recognition of previous research.
Response 6: We agree with this comment. This change can be found Page 6, Line 269 and 271-272. Changes are indicated in red font.
Comments 7: The authors must furnish sufficient references to substantiate their assertions regarding the relationship between ‘Artificial Intelligence Anxiety and Quiet Quitting’ to ensure appropriate attribution and enhance the credibility of their argument.
Response 7: We agree with this comment. This change can be found Page 6, Line 269 and 271-272. Changes are indicated in red font.
Comments 8: In line 257, the authors reference Hamouche et al. (2023)…'; nevertheless, in line 259, they emphasize the same source as 'he puts forward…,' resulting in inconsistency. The writers must specify if 'Baa et al. (2023)' is a single-author or multi-author source and maintain consistency in citation.
Response 8: We agree with this comment. This change can be found Page 7, Line 298. Changes are indicated in red font.
Comments 9: The authors should clearly articulate the specific contributions of uncanny valley theory to this study and how it helps to understand the mediating role of quiet quitting.
Response 9: We agree with this comment. This change can be found Page 16, Line 610-620. Changes are indicated in red font.
Comments 10: The authors may review theses papers and cite them …. to enhance the turnover concept.
Response 10: We agree with this comment. The changes specified by the reviewer have been applied throughout the article. Changes are indicated in red font.
Comments 11: After the first introduction, use the acronym consistently throughout the manuscript. Avoid switching between the full term and the acronym, which can confuse the reader
Response 11: We agree with this comment. The abbreviation Artificial Intelligence, which the reviewer indicated, was applied throughout the article.
Comments 12: The authors must ensure accurate in-text citations to uphold consistency and precision in referencing sources e.g (‘Y.-Y. Wang and Wang, 2022’) should be Wang and Wang, 2022. Authors should ensure same for subsequent ones.
Response 12: We agree with this comment. This change can be found Page 3, Line 140; Page 4, Line 154; Page 4, Line 160; Page 9, Line 378 and Page 14, Line 543. Changes are indicated in red font.
Comments 13: I was expecting the authors to begin their literature review by discussing the relationship between the main independent and dependent variables first, followed by the link between the independent variable and the mediating variable in that orther. Restructuring the literature review in this order will require renumbering the hypotheses, which should also be reflected in the research model.
Response 13: We agree with this comment. The abbreviation Artificial Intelligence, which the reviewer indicated, was applied throughout the article.
Comments 14: The authors should provide the source of their model eg (Authors construct, 2025).
Response 14: We agree with this comment. This change can be found Page 8, Line 334 and Page 13, Line 485. Changes are indicated in red font.
Comments 15: The authors failed to address the research design. It is essential for them to furnish a lucid clarification of the research strategy to enhance the understanding of the study's structure and data collection methods.
Response 15: We agree with this comment. This change can be found Page 8, Line 336-347. Changes are indicated in red font.
Comments 16: The authors assert that their target population is roughly thirty thousand, although they fail to cite a source for this data. For their data to be more credible, they should provide a suitable reference or describe how this estimate was arrived at. Hair et al.'s (2014) guidelines indicate that the optimal sample size should be determined by aspects such as research design, variable count, and analytical methodologies, rather than solely by population size.
Response 16: We agree with this comment. This change can be found Page 8, Line 336-347. Changes are indicated in red font.
Comments 17: The authors effectively cited the origins of their instruments and presented their alpha values. Nonetheless, they did not provide a rationale for the use of these instruments within the context of Turkish SMEs. The authors should elucidate the relevance and validity of these measurements for this particular context.
Response 17: We agree with this comment. This change can be found Pages 8-9, Line 363-374. Changes are indicated in red font.
Comments 18: The authors failed to indicate the specific scale used to measure these items (e.g., strongly disagree to strongly agree). It is important for them to clarify the scale used to ensure transparency in their measurement approach.
Response 18: We agree with this comment. This change can be found Page 9, Line 376-378. Changes are indicated in red font.
Comments 19: The authors failed to address the criteria for participant inclusion and exclusion. It is imperative for them to delineate these criteria to augment the transparency and quality of their sample selection method.
Response 19: We agree with this comment. This change can be found Page 9, Line 395-408. Changes are indicated in red font.
Comments 20: The authors did not specify the sample method, but they mentioned distributing questionnaires to 457 individuals. The writers must delineate the sampling technique employed to elucidate the participant selection process and support the sample's validity.
Response 20: We agree with this comment. This change can be found Page 9, Line 395-408. Changes are indicated in red font.
Comments 21: The authors did not address the statistical software and tools employed for analysis. It is essential for them to delineate the software and analytical techniques utilized to guarantee transparency and repeatability of the study.
Response 21: We agree with this comment. This change can be found Page 9, Line 395-408. Changes are indicated in red font.
Comments 22: I observed that the authors utilized the SMARTPLS tool; however, this did not come up in the methodology or data analysis sections. To promote clarity and transparency, the authors should explicitly articulate their rationale for selecting SMARTPLS and detail its application. In the absence of this, it is challenging to thoroughly evaluate the validity of the results.
Response 22: We agree with this comment. This change can be found Page 9, Line 395-408. Changes are indicated in red font.
Comments 23: The assessment of discriminant validity was briefly mentioned, although the outcomes of the HTMT and Fornell-Larcker tests were not elaborated upon. The authors have to provide a more comprehensive explanation of these data to enhance transparency and bolster the validity of their conclusions. The authors should follow the approach outlined in these papers …
Response 23: We agree with this comment. This change can be found Page 12, Line 464-466 and Line 472-474. Changes are indicated in red font.
Comments 24: … to enhance the structure of their findings and may cite them where appropriate."
Response 24: We agree with this comment. This change can be found Page 12, Line 464-466 and Line 472-474. Changes are indicated in red font.
Comments 25: The manuscript fails to address the mitigation of multicollinearity and common method bias. The authors must discuss how these issues were addressed to guarantee the robustness and validity of their findings.
Response 25: We agree with this comment. This change can be found Page 10, Line 437-440 and Page 11, Line 444-447. Changes are indicated in red font.
Comments 26: The authors presented the R2 and Q2 values but ignored to address their implications. The authors must interpret these values within the framework of their model to explain the explanatory power and predictive significance of their findings.
Response 26: We agree with this comment. This change can be found Page 14, Line 512-514 and Line 516-517. Changes are indicated in red font.
Comments 27: The authors have to present and analyze the effect size (f2) values to elucidate the practical significance of their findings and the robustness of the relationships within the model.
Response 27: We agree with this comment. This change can be found Page 14, Line 518-526. Changes are indicated in red font.
Comments 28: The hypothesis was tested using a bootstrapping resampling of 5,000, not 5.000. The authors must amend this statement for precision.
Response 28: We agree with this comment. This change can be found Page 12, Line 475-481. Changes are indicated in red font.
Comments 29: Simultaneously presenting results for both direct and indirect effects, especially in mediation models, is the conventional method, as it offers a more cohesive and thorough comprehension of the relationships under examination. Nevertheless, the authors did not utilize SPSS to perform a Sobel test for this purpose. This technique is inapplicable due to the utilization of SMARTPLS. The authors should explore utilizing the bootstrapping method, commonly implemented in SMARTPLS for assessing mediation effects.
Response 29: We agree with this comment. This change can be found Page 12, Line 475-481. Changes are indicated in red font.
Comments 30: The authors must specify the hypothesis number to correspond with the recommendations presented in the literature review (i). The hypothesis number should be incorporated into Table 4 for enhanced clarity and consistency.
Response 30: We agree with this comment. This change can be found Page 13, Line 494. Changes are indicated in red font.
Comments 31: The authors should provide separate sections for their discussions, conclusions, theoretical implications, managerial implications, and limitations, along with directions for future research. This would help organize the manuscript more clearly and improve the flow of the content.
Response 31: We agree with this comment. This change can be found Page 16, Line 606-653. Changes are indicated in red font.
Comments 32: The authors should provide separate sections for their discussions, conclusions, theoretical implications, managerial implications, and limitations, along with directions for future research. This would help organize the manuscript more clearly and improve the flow of the content.
Response 32: We agree with this comment. This change can be found Page 16, Line 606-628. Changes are indicated in red font.
Comments 33: The authors acknowledged that the utilization of cross-sectional data was a restriction in their analysis; however, this was not addressed in the methodology section. The authors should acknowledge this restriction in the methodology section to enhance clarity regarding the study design and its possible influence on the findings.
Response 33: We agree with this comment. This change can be found Page 9, Line 400-403. Changes are indicated in red font.
Best Regards
Round 2
Reviewer 2 Report
Comments and Suggestions for Authors
The authors addressed almost all comments with notable ability. Their diligent efforts to enhance the material and meticulous attention to detail during the work's finalization are greatly appreciated. Prior to approval for publication, the authors must address a few overlooked comments.
- The authors did not provide a clear concluding section, despite a request for its submission. The conclusion section should thoroughly discuss the primary findings of the study.
- The authors did not provide clarification on the managerial implications, despite being requested to do so. A section outlining the managerial implications of the findings for practitioners and managers should be included.
Author Response
Dear Reviewer 2
First of all, I would like to express my gratitude for your precious comments and suggestions to improve the article. I took your suggestions into consideration and made the following changes in the article;
Comments 1: The authors did not provide a clear concluding section, despite a request for its submission. The conclusion section should thoroughly discuss the primary findings of the study.
Response 1: We agree with this comment. This change can be found Pages 15-16, Line 597-631. Changes are indicated in red font.
Comments 2: The authors did not provide clarification on the managerial implications, despite being requested to do so. A section outlining the managerial implications of the findings for practitioners and managers should be included.
Response 2: We agree with this comment. This change can be found Pages 16-17, Line 632-672. Changes are indicated in red font.
Best Regards